

# The complete chloroplast genome of the Jerusalem artichoke (*Helianthus tuberosus* L.) and an adaptive evolutionary analysis of the *ycf2* gene

Qiwen Zhong[1,2,3,*], Shipeng Yang[2,3,*], Xuemei Sun[1,2,3], Lihui Wang[2,3] and Yi Li[1]

[1] Northwest Institute of Plateau Biology, Chinese Academy of Sciences, Qinghai Key Laboratory of Qinghai-Tibet Plateau Biological Resources, Xining, Qinghai, China
[2] Agriculture and Forestry Sciences of Qinghai University, Qinghai Key Laboratory of Vegetable Genetics and Physiology, Xining, Qinghai, China
[3] Qinghai University, The Open Project of State Key Laboratory of Plateau Ecology and Agriculture, Xining, Qinghai, China
* These authors contributed equally to this work.

Corresponding author
Yi Li, liyi@nwipb.cas.cn

## ABSTRACT

Jerusalem artichoke (*Helianthus tuberosus* L.) is widely cultivated in Northwest China, and it has become an emerging economic crop that is rapidly developing. Because of its elevated inulin content and high resistance, it is widely used in functional food, inulin processing, feed, and ecological management. In this study, Illumina sequencing technology was utilized to assemble and annotate the complete chloroplast genome sequences of Jerusalem artichoke. The total length was 151,431 bp, including four conserved regions: A pair of reverse repeat regions (IRa 24,568 bp and IRb 24,603 bp), a large single-copy region (83,981 bp), and a small single-copy region (18,279 bp). The genome had a total of 115 genes, with 19 present in the reverse direction in the IR region. A total of 36 simple sequence repeats (SSRs) were identified in the coding and non-coding regions, most of which were biased toward A/T bases. A total of 32 SSRs were distributed in the non-coding regions. A comparative analysis of the chloroplast genome sequence of the Jerusalem artichoke and other species of the composite family revealed that the chloroplast genome sequences of plants of the composite family were highly conserved. Differences were observed in 24 gene loci in the coding region, with the degree of differentiation of the *ycf2* gene being the most obvious. A phylogenetic analysis showed that *H. petiolaris subsp. fallax* had the closest relationship with Jerusalem artichoke, both members of the *Helianthus* genus. Selective locus detection of the *ycf2* gene in eight species of the composite family was performed to explore adaptive evolution traits of the *ycf2* gene in Jerusalem artichoke. The results show that there are significant and extremely significant positive selection sites at the 1239N and 1518R loci, respectively, indicating that the *ycf2* gene has been subject to adaptive evolution. Insights from our assessment of the complete chloroplast genome sequences of Jerusalem artichoke will aid in the in-depth study of the evolutionary relationship of the composite family and provide significant sequencing information for the genetic improvement of Jerusalem artichoke.

# INTRODUCTION

Jerusalem artichoke (*Helianthus tuberosus* L.) is a species of the composite family native to North America, primarily distributed in the temperate zone of 40–55 °C north latitude and the temperate region with the approximate similar latitude in the southern hemisphere. Jerusalem artichoke was introduced to China via Europe in the 17th century. It has been grown on a small scale as a pickled vegetable in various regions of China. Jerusalem artichoke is highly resistant and can be grown in saline, alkaline, dry, and low temperature conditions. Therefore, it is widely cultivated in various regions of China, especially in the Qinghai plateau in recent years. To date, most research on Jerusalem artichoke has focused on ecological management, feed research and development, and the processing of inulin products. Studies centered on the improvement of saline land in the Songnen Plain have recognized Jerusalem artichoke as an excellent improved crop, which has already been initially grown in saline-alkali grassland (*Yan, Li & Wang, 2008*). The aboveground part of Jerusalem artichoke is tall, making it an easily accessible source of animal feed. Furthermore, its leaves are particularly nutritious compared with other feed ingredients, being rich in lysine and methionine, and having a dry matter content of protein as high as 20%, of which 5–6% corresponds to lysine, an essential amino acid (*Rawate & Hill, 1985*). Jerusalem artichoke also utilizes fructan as a source of carbon, instead of starch, as most crops. Fructan can be processed or modified, providing the raw materials for the production of bioethanol, paper, and healthcare products (*Saengkanuk et al., 2011*; *Wang et al., 2015*; *Wyse, Young & Jones, 2017*).

The composite family is the largest group of dicotyledonous chrysanthemums, encompassing 25,000–30,000 species distributed throughout the world. A total of 52 species and a large number of subspecies have been recognized in the *Helianthus* genus, including Jerusalem artichoke. The morphology of these plants is complex and diverse, leading to difficulties in identification and evolutionary analysis. Jerusalem artichoke is a hexaploid species ($2n = 6x = 102$), which reproduces primarily through vegetative propagation by tubers (*Baldini et al., 2004*). The evolutionary assessment of this plant is controversial, with its ancestral species remaining uncertain. Hybridization experiments between Jerusalem artichoke and *H. annuus* L. have confirmed homologous genes between these species. It is generally believed that the chromosome number of triploid hybrid (AAB) in Jerusalem artichoke has doubled. Moreover, cytogenetic studies have demonstrated that two of the three genomes of Jerusalem artichoke are homologous (*Atlagić, Dozet & Škorić, 1993*; *Kostoff, 1934*, *1939*). The diploid ($2n = 2x = 34$) B genome is provided by the immediate ancestor of *H. annuus* L., while the autotetraploid ($2n = 4x = 68$) A genome is provided by the crop in the composite family (*Bock et al., 2014*; *Heiser & Smith, 1964*; *Heiser et al., 1969*). *Helianthus hirsutus* is regarded as the most likely tetraploid ancestor (*Bock et al., 2014*), while *H. grosseserratus*, and *H. giganteus* are viewed as the most likely diploid ancestors. The sequencing of related species using partial

mitochondrial genomes, as well as 35S and 5S ribosomal DNA, has shown the origin of Jerusalem artichoke to be very rich and probably linked to the hybridization of tetraploid Hairy *H. annuus* L. and diploid Sawtooth *H. annuus* L. (*Bock et al., 2014*; *Timme, Simpson & Linder, 2007*). With the development of high-throughput sequencing technology, chloroplast phylogenetic genome evaluation has become a hot topic in the evolutionary research of plants in recent years. Plenty of phylogenetic information is contained in the chloroplast genome, providing a broad data platform for the study of phyletic evolution, and thereby verifying and extending the results of previous studies. The chloroplast genome sequencing of eight *Helianthus* species has been completed. However, this aspect remains unexplored concerning Jerusalem artichoke.

Thus, in this study, we report the complete chloroplast genome sequencing, assembly and comparative analysis of Jerusalem artichoke. This data will help elucidate the evolutionary history of Jerusalem artichoke and its phylogenetic position in the composite family. In addition, it will lay a foundation for further studies of population genetics and other molecular aspects of Jerusalem artichoke based on chloroplast DNA sequencing.

## MATERIALS AND METHODS

### Samples and genome sequencing

Fresh tender leaves of Jerusalem artichoke were obtained from the experimental base of the Qinghai Academy of Agricultural and Forestry Sciences (N36°43′51, E101°45′24). Chloroplast DNA was extracted through an improved high-throughput chloroplast genome extraction method (*Shi et al., 2012*). Illumina HiSeq PE150 paired-end sequencing technology was used to establish the library for sequencing. The library was of the DNA small fragment type with 400, 150 bp read length with the average depth was 100×.

### Chloroplast genome assembly and annotation

FastQC was used for the quality filtering of clean data. SOAPdenovo software was used for pre-assembly (*Lee & Lee, 1995*), while SPAdes v3.6.2 (http://bioinf.spbau.ru/spades) was used for sequence assembly (*Bankevich et al., 2012*). The sequence of the chloroplast genome of *H. annuus* L. was used as a reference to determine the location of the chloroplast genome. Gapcloser (*Luo et al., 2012*) and GapFiller (*Boetzer & Pirovano, 2012*) software for repairing gaps, and PrInSeS-G was then used for sequence correction. DOGMA software (http://dogma.ccbb.utexas.edu/) (*Wyman, Jansen & Boore, 2004*) was used for annotation. The above program uses default parameters. The gene region and protein coding sequence were manually adjusted according to the initiation codon and termination codon sequences. tRNA was entered into tRNAscan-SE (http://lowelab.ucsc.edu/tRNAscan-SE/) for annotation (*Lowe & Chan, 2016*). rRNA was submitted to the RNAmmer 1.2 Server (http://www.cbs.dtu.dk/services/RNAmmer/) for prediction. The resulting sequence information and annotation results were submitted to Genebank, with the sequence number of MG696658. The Organellar Genome DRAW software (http://ogdraw.mpimp-golm.mpg.de/index.shtml) (*Lohse et al., 2013*) was used to render a complete circular chloroplast genome map.

## Repeats and SSRs analysis

The chloroplast genome was entered into REPuter (*Kurtz et al., 2001*) to identify forward and reverse repeat sequences. Simple sequence repeats (SSRs) were identified by MIcroSAtellite software based on a perl script (http://pgrc.ipk-gatersleben.de/misa/). The number of repeats from mononucleotide to hexanucleotide was set to 10, 5, 4, 3, 3, and 3.

## Comparative analysis of different *Asteraceae* plastomes

The LAGAN model in the mVISTA software (*Frazer et al., 2004*) was used to perform a comparative analysis of the chloroplast genome of Jerusalem artichoke with *Carthamus tinctorius* (KX822074.1), *Ageratina adenophora* (JF826503.1), *Guizotia abyssinica* (EU549769.1). *Lactuca sativa* (NC_007578.1), *H. argophyllus* (KU314500.1), *H. debilis* (KU312928.1), and *H. petiolaris subsp. fallax* (KU295560.1). After screening for the quality of the original chloroplast genome data of Jerusalem artichoke, the final constructed sequence (the gene sequence extracted from the annotation) and the established chloroplast genome of 15 plant species were compared by Blast+ (ftp://ftp.ncbi.nlm.nih.gov/blast/executables/blast+/LATEST/). HomBlocks (*Bi et al., 2018*) was used to construct a Circos map (http://circos.ca/) to find the direction, relative position and link color of the genes. This was then standardized according to the length of all the alignment regions. Coloring was performed in accordance with the long, medium, relative short, and short sequence lengths (pink, orange, green, and blue, respectively). COBALT (https://www.ncbi.nlm.nih.gov/tools/cobalt/cobalt.cgi?CMD=Web) was utilized to compare the differential protein sequence *ycf2*. HomBlocks and COBALT use default parameters.

## Phylogenetic analysis

The following 15 species of the composite family were used for the phylogenetic analysis of Jerusalem artichoke: *Ageratina adenophora* (JF826503.1), *Carthamus tinctorius* (KX822074.1), *G. abyssinica* (NC_010601.1), *Jacobaea vulgaris* (NC_015543.1), *L. sativa* (NC_007578.1), *H. annuus* (NC_007977.1), *H. petiolaris subsp. fallax* (KU295560.1), *H. argophyllus* (KU314500.1), *H. debilis* (KU312928.1), *H. annuus cultivar line HA383* (DQ383815.1), *H. petiolaris* (KU310904.1), *H. praecox* (KU308401.1), *H. annuus subsp. Texanus* (KU306406.1), *Mikania micrantha* (NC_031833.1), and *Taraxacum mongolicum* (NC_031396.1). MAFFT 7.388 (*Katoh, Rozewicki & Yamada, 2017*) was used to compare 16 chloroplast genome sequences. A phylogenetic tree was constructed with the methods of maximum-likelihood and Bayesian, respectively. The GTRGAMMAI model was used in the ML Tree, and RAxML v8.1.24 (*Stamatakis, 2014*) was used to construct the tree. Parameters were set to search for 30 repeats, and the tree with the maximum likelihood value was used. In addition, Bootstrap was set to run 1,000 times to calculate the support of each branch. To build the Bayesian tree, the nucleotide substitution model GTR+I+G in Bayesian analysis was selected according to BIC in the jModelTest 2.1.7 software (*Darriba et al., 2012*). MrBayes 3.2 (*Ronquist et al., 2012*) was used for calculations, employing the Markov chain Monte Carlo methodology. Four Markov chains were initialized at the same time. The random tree was marked as the initial tree, and one was saved every 500 trees for a total of 5,000,000 trees. The first 20% of the burn-in

trees were discarded. The remaining trees were used to calculate the posterior probability of the consistent tree and each branch.

## Adaptive evolution traits

The ratio ($\omega$) of the non-synonymous substitution (dN) to the synonymous substitution (dS) of nucleotides is used in most adaptive evolution studies to measure the selection pressure at the nucleic acid or protein level. In addition, the selection pressure is considered to hinder or promote its role in the process of non-synonymous replacement fixation. The positive selection model (M2a, M8) and the control model (M1a, M7, M8a) provided by EasyCodeML software were used to conduct the adaptive evolution analysis in the loci (*Gao et al., 2019*). The locus model was used to assume that there were different selection pressures at different loci. In other words, the $\omega$ values were different, but there was no difference in the different branches of the phylogenetic tree. This model was primarily used to detect the existence of positive selection ($\omega > 1$) and negative selection ($\omega < 1$) loci in the *ycf2* gene. Three pairs of comparison models were M1a (near neutral) and M2a (selection), M0 (single ratio) and M3 (discrete), M7 (beta) and M8 (beta & $\omega$) in this study. The former is a zero hypothesis, and the latter is an alternative hypothesis. Models M0 (single ratio) to M3 (discrete) were used to detect different $\omega$ values at each point rather than detecting positive selection loci. PAMLx V1.3.1 was used to perform the likelihood ratio test (LRT) in three pairs of models (*Xu & Yang, 2013*). Positive selection loci were tested by comparing the significance of the differences between the models. $\chi^2$ distribution was used as the significance test under the condition of relative degrees of freedom (the difference between the number of two models).

# RESULTS

## Genome organization and gene features

The chloroplast genome of Jerusalem artichoke had a total length of 151,431 bp. The genome was composed of four parts: A pair of reverse repeat regions, IRa (24,568 bp) and IRb (24,603 bp), separated by a large single-copy region LSC (83,981 bp) and a small single-copy region SSC (18,279 bp) (Fig. 1). Genes in the coding regions accounted for 55.45% of the genome, including protein-coding genes, tRNA genes, and rRNA genes. The chloroplast genome of Jerusalem artichoke had a total guanine-cytosine content (GC content) of 37.6%, with GC in the IR region corresponding to 43.2%, and GC in the LSC and SSC regions being 35.6% and 31.3%, respectively. The chloroplast genome of Jerusalem artichoke contained 115 genes, including 84 protein-coding genes CDS, 27 tRNA genes and four rRNA genes distributed in the IR region. Furthermore, this region encompassed 19 inverse genes, including eight CDS genes (*ycf2*, *ndhB*, *rps7*, *rps12*, *ycf15*, *ycf1*, *rpl2*, and *rpl23*), seven tRNA genes, and four rRNA genes. The 115 genes contained 60 Protein synthesis and DNA replication genes, 44 Photosynthesis genes, six Miscellaneous group genes and five pseudogenes of unknown function (Table 1). In the chloroplast genome of Jerusalem artichoke, 16 intron-containing genes were annotated, 11 of which were protein-encoding and five were tRNA genes. Of the 16 intron genes, the intron sequence in *trnK-UUU* was the longest (2,528 bp), while the intron in the

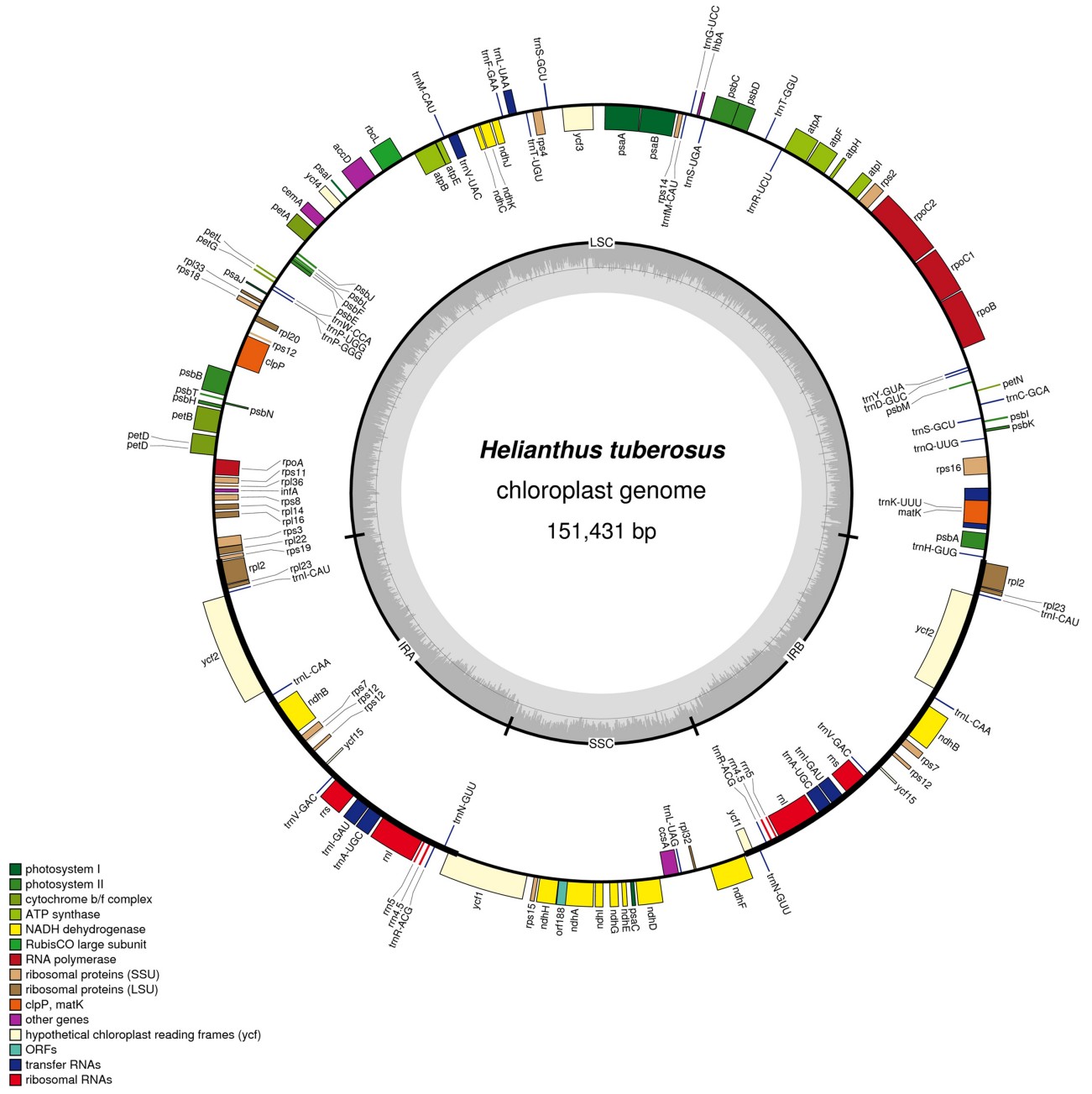

**Figure 1** **Gene map of the *Helianthus tuberosus* L. chloroplast genome.** Genes drawn outside of the circle are transcribed counter-clockwise, while genes shown on the inside of the circle are transcribed clockwise. Genes belonging to different functional groups are color-coded. The darker gray in the inner circle indicates GC content, while the lighter gray corresponds to AT content.

trnL-UAA gene was the smallest (436 bp). There were two introns in the *clpP*, *ycf3*, and *rps12* genes, whereas the other genes contained only one intron (Table 2). Since Bock et al. have sequenced the Jerusalem artichoke plastid genome, based on this, we performed a detailed comparison (NCBI accession: NC_023112), and the sequencing results in this study (NCBI accession: MG696658), which are shown by the results of BRIG (Fig. 2). The result of this sequencing indicate that there are 384 bp more than in NC023112, and

**Table 1 List of genes in the chloroplast genome of *Helianthus tuberosus* L.**

| | Groups of genes | Names of genes |
|---|---|---|
| Protein synthesis and DNA replication | Ribosomal RNAs | *16S r RNA(2×), 23S r RNA(2×), 4.5S r RNA(2×), 5S r RNA(2×)* |
| | Transfer RNAs | *trnQ-TTG, trnL-TAG, trnD-GTC, trnS-GGA, trnE-TTC, trnS-GCT, trnY-GTA, trnV-GAC, trnP-TGG, trnH-GTG, trnF-GAA, trnN-GTT, trnT-TGT, trnW-CCA, trnS-TGA, trnV-GAC, trnL-CAA(2×), trnM-CAT(2×), trnC-GCA, trnI-CAT, trnT-GGT, trnI-CAT, trnR-ACG, trnN-GTT, trnR-TCT, trnR-ACG, trnG-GCC* |
| | Ribosomal protein small subunit | *rps7, rps14,rps12, rps2, rps4, rps12, rps7, rps11, rps16, rps12, rps19 (2×), rps3, rps15, rps8, rps19* |
| | Ribosomal protein large subunit | *rpl14, rpl23, rpl36, rpl2, rpl20, rpl2, rpl32, rpl16, rpl33, rpl23, rpl22* |
| | Subunits of RNA polymerase | *rpoB, rpoC(2×), rpoA* |
| Photosynthesis | Photosystem I | *psaC, psaA, psaB, psaI, psaJ* |
| | Photosystem II | *psbZ, psbK, psbB, psbI, psbF, psbN, psbL, psbJ, psbC, psbE, psbM, psbH, psbA, psbD, psbT* |
| | Cytochrome b/f complex | *petA, petD, petL, petB, petG, petN* |
| | ATP synthase | *atpE, atpH, atpA, atpI, atpF, atpB* |
| | NADH-dehydrogenase | *ndhJ, ndhA, ndhK(2×), ndhG, ndhI, ndhB(2×), ndhH, ndhE, ndhD, ndhC, ndhF* |
| | Large subunit Rubisco | *rbcL* |
| Miscellaneous group | Translation initiation factor IF-1 | *infA* |
| | Acetyl-CoA carboxylase | *accD* |
| | Cytochrome c biogenesis | *ccsA(2×)* |
| | Maturase | *matK* |
| | ATP-dependent protease | *clpP* |
| | Inner membrane protein | *cemA* |
| Pseudogenes of unknown function | Conserved hypothetical chloroplast open reading frame | *ycf15(4×), ycf4, ycf3, ycf1(2×), ycf2(2×)* |

there are partial base differences in 15 genes: *ccsA, atpB, clpP, ndhB, ndhH, ndhI, petA, petD, rpl2, rpoC1, rpoC2, rps12, rps16, ycf1*, and *ycf2*, with multiple differences in *clpP* and *rpoC1* (Table 3).

## Repeats and SSRs analysis

The distribution of chloroplast simple sequence repeat (cpSSR) in Jerusalem artichoke was analyzed, revealing 36 different SSR loci in its chloroplast genome. Among them, 32 SSR were composed of A or T, two were composed of C, and only one was composed of G, indicating that the chloroplast genomic SSR of Jerusalem artichoke are biased toward A/T bases (Fig. 3). An assessment of the SSR distribution identified 32 SSR in the non-coding region of the chloroplast genome. The non-coding region primarily includes an intergenic spacer and introns, accounting for 68% and 20% of the distribution, respectively. In the coding region, SSR are only found in the *rpoC2, cemA*, and *ycf1* genes.

## Comparative analysis of different composite chloroplast

A comparative analysis with the plastomes of other species of the composite family revealed only small differences in plastome size and composition in comparison to that of

**Table 2 Characteristics of genes including introns and exons in the chloroplast genome of *Helianthus tuberosus* L.**

| Gene | Region | Exon I (bp) | Intron I (bp) | Exon II (bp) | Intron II (bp) | Exon III (bp) |
|------|--------|------|------|------|------|------|
| *trnK-UUU* | LSC | 51 | 2,528 | 36 | | |
| *rps16* | LSC | 29 | 864 | 226 | | |
| *rpoC1* | LSC | 431 | 733 | 1,727 | | |
| *atpF* | LSC | 144 | 714 | 391 | | |
| *ycf3* | LSC | 152 | 746 | 229 | 700 | 123 |
| *trnL-UAA* | LSC | 36 | 436 | 49 | | |
| *trnV-UAC* | LSC | 36 | 574 | 37 | | |
| *clpP* | LSC | 68 | 792 | 290 | 624 | 227 |
| *petB* | LSC | 5 | 775 | 641 | | |
| *petD* | LSC | 8 | 712 | 473 | | |
| *rpl2* | LSC | 392 | 663 | 434 | | |
| *ndhB* | IR | 755 | 671 | 776 | | |
| *trnI-GAU* | IR | 41 | 776 | 34 | | |
| *trnA-UGC* | IR | 37 | 822 | 34 | | |
| *ndhA* | SSC | 552 | 1,095 | 538 | | |
| *rps12* | LSC-IR | 113 | | 230 | | 29 |

Jerusalem artichoke (Table 4). There were very few inconsistencies in the types and number of chloroplast genes in several species of the composite family, and the types and number were very conserved. The total size chloroplast genome of Jerusalem artichoke ranked 5th in the aligned genomes of the eight chloroplast genomes of the composite family. The variation in the length of the sequence may be caused by the difference in length between the LSC and IR regions. The chloroplast genome size of eight crops of the composite family was approximately 150 kb, with a GC content of approximately 37.5%. The number of protein-coding genes ranged between 79 and 89. All of these genomes had four rRNA-coding genes and 20–30 tRNA-coding genes. The plastome of Jerusalem artichoke was 327 bp longer than that of *H. petiolaris subsp. fallax* (a crop in the same genus), primarily in the LSC region. In addition, it had five more protein-coding genes than that of *H. petiolaris subsp. fallax*, with no difference in the number of rRNA- and tRNA-coding genes.

The genomic sequences of eight composite species were analyzed by the mVISTA software, detecting the variations of the sequences (Fig. 4). The results showed there was less variation between Jerusalem artichoke, *H. petiolaris subsp. fallax* and *H. debilis* and *H. argophyllus*. Compared with *Ageratina adenophora*, a partial structure was lacking in the Jerusalem artichoke.

Based on the results of mVISTA, a systematic comparative analysis was performed in a coding region with small variation amplitude (*Doorduin et al., 2011*). As shown in Fig. 5, there were differences among eight species of the composite family in the following 24 gene loci: *trnN-GUU, trnR-ACG, trnA-UGU, ycf68, trnL-GAU, trnV-GAC,*

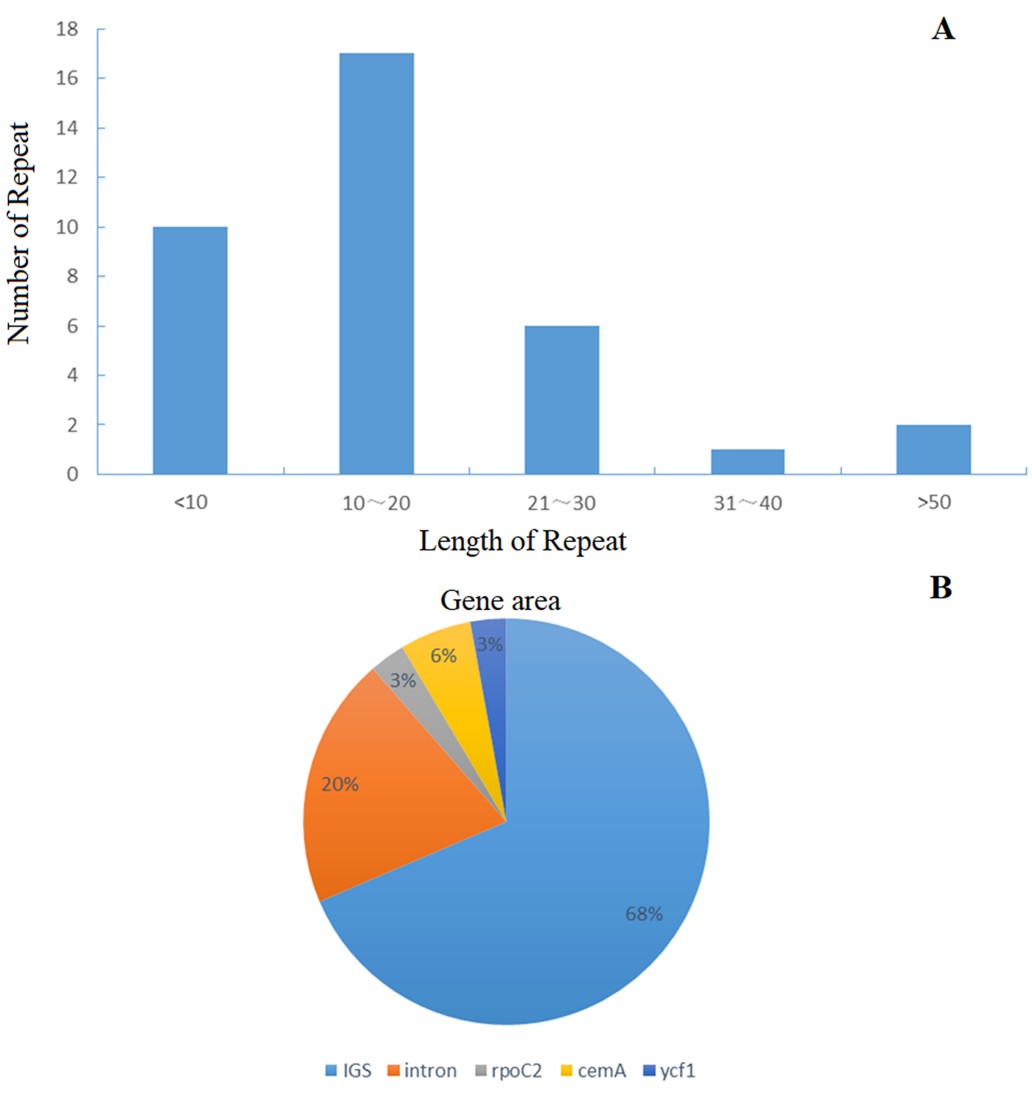

**Figure 2 Distribution frequency in *Helianthus tuberosus* L. cp genome.** (A) The frequency of repeats, length of repeats; Number of repeats. (B) The percentage distribution of gene area.

*ycf15*, *rps7*, *ndhB*, *trnL-CAA*, *ycf2*, *trnL-CAU*, *rpl23*, *rpl2*, *rps19*, *rps12*, *rpl20*, *rps18*, *rpl33*, *trnP-UGG*, *petL*, *trnG-UCC*, *trnS-GCU*, and *trnC-GCA*. The discovery of these differential genes provides valuable phylogenetic information for the further evaluation of the composite family.

In many studies, the *ycf2* gene has become an alternative choice for the assessment of plant sequence variation and phylogenetic evolution. Our results showed that the *ycf2* gene segment had a large deletion and inconsistency. The *ycf2* gene of Jerusalem artichoke and seven other composite species was compared. Four species of the genus *Helianthus* had 152 amino acid sequence deletions of the *ycf2* gene in the segment 308–460 (Fig. 6). In addition, only *H.s petiolaris* had 12 amino acid sequence deletions in the segment 1,524–1,536 among four *Helianthus* species. There were 12 amino acid sequence deletions

**Table 3 Comparison of chloroplast and plastid differential genes in *Helianthus tuberosus* L.**

| Gene | NCBI accession | Difference site | | | | Difference position and base |
|------|------|------|------|------|------|------|
| | | T | C | A | G | |
| *ccsA* | MG696658 | 36.8 | 15.6 | 31.6 | 16.0 | |
| | NC023112 | 36.9 | 15.5 | 31.6 | 16.0 | 579T |
| *atpB* | MG696658 | 36.8 | 15.6 | 31.6 | 16.0 | |
| | NC023112 | 36.9 | 15.5 | 31.6 | 16.0 | 348G |
| *clpP* | MG696658 | 28.9 | 18.0 | 28.6 | 24.5 | 361–363 null |
| | NC023112 | 29.1 | 18.1 | 28.3 | 24.5 | 362G/363C/70,361T |
| *ndhB* | MG696658 | 34.7 | 19.6 | 27.6 | 18.0 | |
| | NC023112 | 34.8 | 19.5 | 27.9 | 17.8 | 778–819 null |
| *ndhH* | MG696658 | 31.0 | 15.2 | 30.9 | 22.9 | |
| | NC023112 | 30.9 | 15.2 | 30.9 | 23.0 | 822G |
| *ndhI* | MG696658 | 34.1 | 16.2 | 31.5 | 18.2 | |
| | NC023112 | 33.9 | 16.4 | 31.5 | 18.2 | 433C |
| *petA* | MG696658 | 28.9 | 19.3 | 30.8 | 21.0 | |
| | NC023112 | 28.9 | 19.3 | 30.7 | 21.1 | 705G |
| *petD* | MG696658 | 32.9 | 19.0 | 27.5 | 20.5 | |
| | NC023112 | 32.9 | 19.0 | 27.7 | 20.3 | 9A |
| *rpl2* | MG696658 | 22.9 | 18.2 | 33.5 | 25.4 | |
| | NC023112 | 22.9 | 18.3 | 33.5 | 25.3 | 392–394 null |
| *rpoC1* | MG696658 | 30.0 | 16.9 | 32.4 | 20.7 | 2–22 null |
| | NC023112 | 30.0 | 16.9 | 32.4 | 20.7 | 4, 5, 8, 10, 11, 22A/3, 6, 9, 12, G/7, 17, 20, C/2, 13, 14, 15, 16, 18, 19, 21T. |
| *rpoC2* | MG696658 | 29.4 | 17.9 | 32.5 | 20.2 | |
| | NC023112 | 29.4 | 17.9 | 32.6 | 20.2 | |
| *rps12* | MG696658 | 23.7 | 21.3 | 33.1 | 21.9 | 347 null |
| | NC023112 | 24.6 | 21.6 | 30.8 | 23.0 | 346, 356A/347, 349, 351, 354G, 352T/358-376 null |
| *rps16* | MG696658 | 28.5 | 17.2 | 33.0 | 21.3 | |
| | NC023112 | 28.6 | 16.5 | 33.7 | 21.2 | 43–54 null |
| *ycf1* | MG696658 | 30.6 | 14.2 | 39.6 | 15.6 | |
| | NC023112 | 30.6 | 14.2 | 39.7 | 15.5 | 1A. 2–4 null |
| *ycf2* | MG696658 | 31.1 | 18.5 | 31.2 | 19.2 | |
| | NC023112 | 31.1 | 18.5 | 31.2 | 19.1 | 4,562–4,597 null |

in segment 1,641–1,653 of *Ageratina adenophora* and *L. sativa*, as well as in the segment 1,641–1,664 of *G. abyssinica*. In addition, there were some amino acid site differences. Lastly, the greatest similarity was observed between the *ycf2* genes of Jerusalem artichoke and *H. petiolaris subsp. fallax*, with the exception of the presence of five additional amino acids in the start of *ycf2* in the Jerusalem artichoke plastome.

## Phylogenetic analysis

To assess the phylogenetic relationships of Jerusalem artichoke, the chloroplast genomes of 15 species of the composite family were compared globally. *Jacobaea vulgaris* was
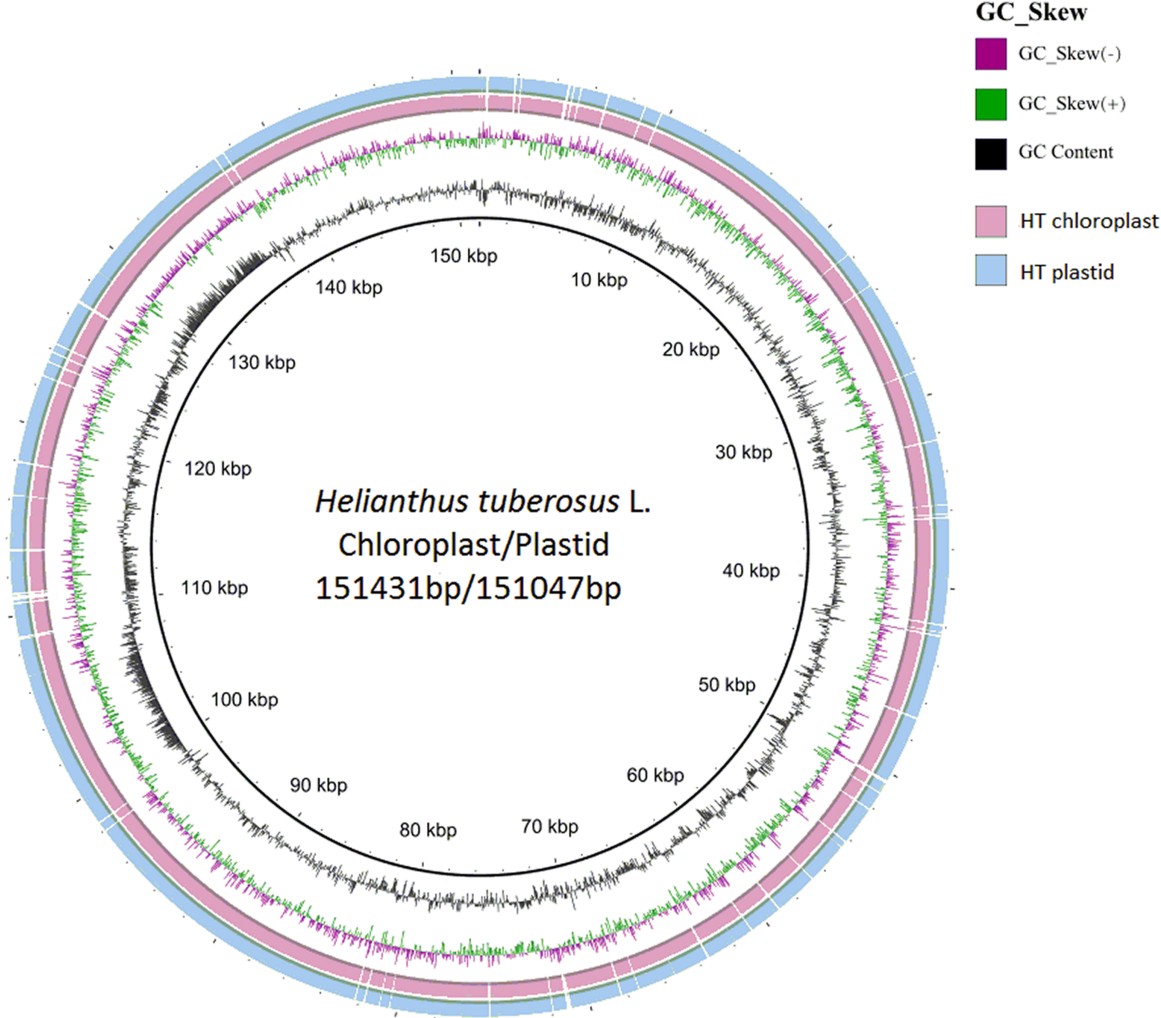

**Figure 3 Compared *Helianthus tuberosus* L. chloroplast and plastid genome use BRIG.**

**Table 4 Comparison of cp genomes among eight composite species.**

| Species | Size (bp) | | | | G+C (%) | Total number of genes | | | GeneBank accessions |
|---------|-----------|-----|-----|-----|---------|------------------------|-------|-------|---------------------|
| | Total | LSC | IR | SSC | | Protein-coding genes | rRNAs | tRNAs | |
| *Carthamus tinctorius* | 153,675 | 83,606 | 25,407 | 19,156 | 37.4 | 89 | 4 | 30 | KX822074 |
| *Ageratina adenophora* | 150,689 | 84,815 | 23,755 | 18,358 | 37.5 | 80 | 4 | 28 | JF826503 |
| *Guizotia abyssinica* | 150,689 | 82,855 | 24,777 | 18,277 | 37.3 | 79 | 4 | 29 | HQ234669 |
| *Lactuca sativa* | 152,772 | 84,105 | 25,034 | 18,599 | 37.5 | 78 | 4 | 20 | DQ383816 |
| *Helianthus tuberosus* | 151,431 | 83,981 | 24,568 | 18,279 | 37.6 | 84 | 4 | 27 | MG696658 |
| *Helianthus argophyllus* | 151,862 | 83,845 | 24,588 | 18,149 | 37.6 | 80 | 4 | 27 | KU314500 |
| *Helianthus debilis* | 151,678 | 83,799 | 24,502 | 18,121 | 37.6 | 82 | 4 | 27 | KU312928 |
| *Helianthus petiolaris subsp. fallax* | 151,104 | 83,530 | 24,633 | 18,308 | 37.6 | 79 | 4 | 27 | KU295560 |

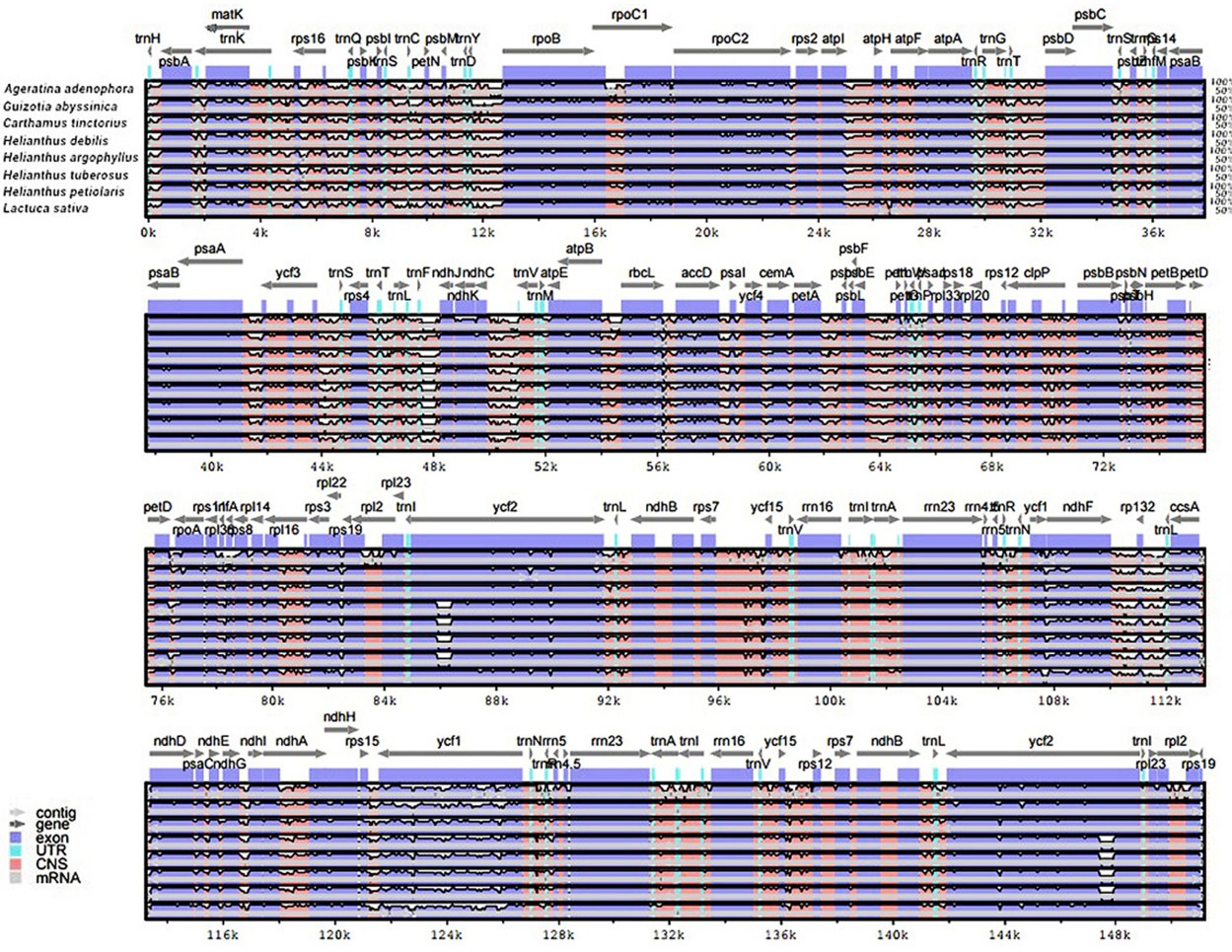

**Figure 4 Percent identity plot for the comparison of eight composite chloroplast genomes.** The whole chloroplast genome was divided into four parts, and the gene names are displayed in sequence on the top line of each part (arrows indicate the transcriptional direction). The sequence similarity of the alignment region of Jerusalem artichoke and seven other species is shown as the filling color in each black stripe. The x-axis indicates the position of the chloroplast genome at a certain site, and the y-axis indicates the average sequence identity percentage (50–100%) with Jerusalem artichoke on the position of a species at a certain position (50–100%). The coding sequences (exons), rRNA, tRNA and the conserved non-coding sequences (CNS) in the genomic region are represented with different colors.

used as an outgroup, and then RAxML and Bayesian evolutionary trees were constructed, respectively. The resulting phylogenetic trees constructed by the two methods shared the same topological structure (Fig. 7). All the species in the composite family formed three highly supported evolutionary clades: members of the genus *Helianthus* are included in the first clade, including some *H. annuus* L. species, subspecies and Jerusalem artichoke, as well as *Eupatorieae* and *Millerieae*. On the evolutionary subclade of the genus *Helianthus*, Jerusalem artichoke and *H. petiolaris subps. fanax* are in the closest relationship. The common node bootstrap is fully resolved. *Lactuca sativa* and *T. officinale* of the Crepidinae are contained in the second clade, while *J. vulgaris* is clustered alone in the Senecioninae.

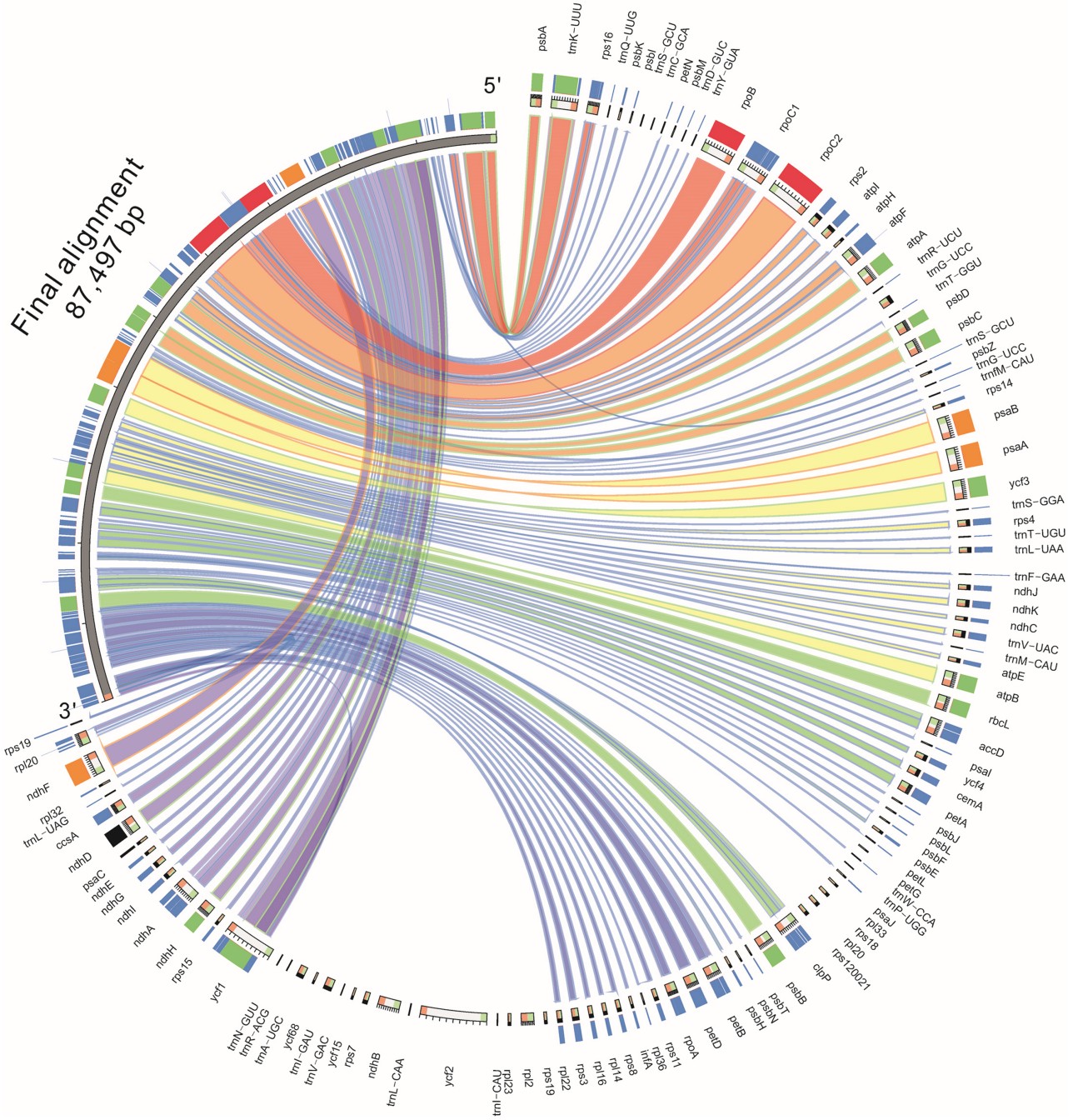

**Figure 5 Comparison of the similarity of chloroplast genomes between Jerusalem artichoke and seven other species of crops in the composite family.**

## Estimation of the positive selection loci of the *ycf2* gene in eight species of the composite family

EasyCodeML v1.2 and paml X1.3 were used to calculate the logarithmic likelihood value (InL) and parameter evaluation for the complete sequence data set of the *ycf2* coding
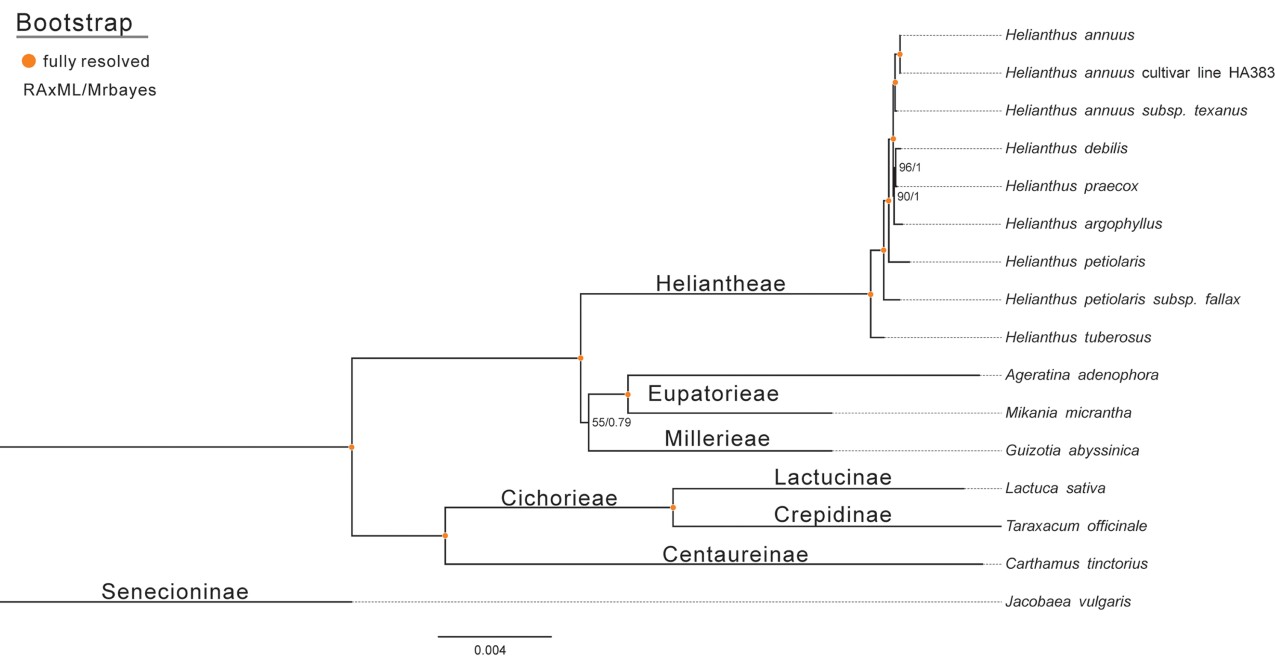

**Figure 6 Comparison of the *ycf2* gene sequence in chloroplast genomes between Jerusalem artichoke and seven other species of crops in the composite family.** The white vacancy corresponds to the missing amino acid sequence.

**Figure 7 Molecular phylogenetic tree of 16 composite species based on a neighbor joining analysis.** Numbers above and below nodes are bootstrap support values 50%.

region of eight species in the composite family. In the locus model, ω > 1 was allowed in the models M3 (discrete), M2a (selection), and M8 (beta & ω) to assume that the corresponding zero hypothetical models were the M1a (near neutral) model, M0 (one-ratio) model and M7 (beta) model. The M3, M2a, and M8 models were significantly superior to their corresponding hypothetical models M0, M1a, M7, and M8a ($p < 0.01$), indicating that there were differences in the selection pressure among the points. After LRT testing, it was found that both M7 vs. M8 and M8a vs. M8 were more consistent with the analyzed data than their original hypothetical models (Table 5), and their original hypothetical models were rejected at a significant level of $p = 0.01$. A consistent positive selection locus, 1239N and 1518R, was found in models M2a and M8, respectively, at 95% and 99% levels calculated by Naïve empirical bayes (NEB) (Table 6). There was one positive selection locus 1518R in the M2a model and two positive selection loci 1239N and 1518R in the M8 model according to a Bayes Empirical Bayes analysis. Overall, the

**Table 5 Likelihood ratio statistics of positive selection models against their null models ($2\Delta \ln L$).**

| Comparison between models | $2\Delta$ lnL | d.f. | p-value |
|---|---|---|---|
| M0 vs. M3 | 15.2245 | 4 | 0.0043 < 0.01 |
| M1a vs. M2a | 13.5353 | 2 | 0.0012 < 0.01 |
| M7 vs. M8 | 15.0177 | 2 | 0.0005 < 0.01 |
| M8a vs. M8 | 13.5241 | 1 | 0.0002 < 0.01 |

**Table 6 Positive selective amino acid loci and parameter estimation in *ycf2* of eight species in the compositae family species.**

| Models | Np | lnL | Estimates of parameters | Positive sites (NEB) | Positive sites (BEB) |
|---|---|---|---|---|---|
| M0 (one-ratio) | 15 | −9,464.31 | $\omega = 0.93903$ | Not allowed | Not allowed |
| M3 (Discrete) | 19 | −9,456.70 | $p_0 = 0.00005$, $\omega_0 = 0.07668$<br>$p_1 = 0.99613$, $\omega_1 = 0.86440$<br>$p_2 = 0.00382$, $\omega_2 = 43.87141$ | 1125W 0.602<br>1238G 0.779<br>1239N 0.980*<br>1476F 0.649<br>1518R 0.992** | Not allowed |
| M1a (Near neutral) | 16 | −9,463.47 | $p_0 = 0.20671$, $\omega_0 = 0$<br>$p_1 = 0.79329$, $\omega_1 = 1$ | Not allowed | Not allowed |
| M2a (Selection) | 18 | −9,456.70 | $p_0 = 0.98950$, $\omega_0 = 0.86336$<br>$p_1 = 0.00668$, $\omega_1 = 1$<br>$p_2 = 0.00382$, $\omega_2 = 43.84482$ | 1125W 0.602<br>1238G 0.779<br>1239N 0.980*<br>1476F 0.649<br>1518R 0.992** | 331I 0.726<br>662K 0.727<br>1125W 0.677<br>1238G 0.770<br>1239N 0.940<br>1476F 0.759<br>1518R 0.950* |
| M7 (beta) | 16 | −9,464.36 | $p = 0.50360$, $q = 0.00500$ | Not allowed | Not allowed |
| M8 (beta & $\omega$) | 18 | −9,460.27 | $p_0 = 0.66725$, $p = 0.00500$<br>$p_1 = 0.33275$, $q = 1.20677$<br>$\omega = 2.95373$ | 1125W 0.600<br>1238G 0.778<br>1239N 0.980*<br>1476F 0.647<br>1518R 0.991** | 331I 0.882<br>662K 0.823<br>1095S 0.526<br>1125W 0.774<br>1238G 0.851<br>1239N 0.965*<br>1476F 0.844<br>1518R 0.971* |
| M8a (beta & $\omega$ = 1) | 17 | −9,463.50 | $p_0 = 0.21119$, $p = 3.03780$<br>$p_1 = 0.78881$, $q = 1.57211$<br>$\omega = 1$ | Not allowed | Not allowed |

**Note:**
Positively selected sites (*$p > 95\%$; **$p > 99\%$).

posterior probabilities of 1239N and 1518R in the NEB analysis of the M2a and M8 models were greater than 95% and 99%, respectively. Currently, this type of gene has substantial potential for application and diverse functions in the field of plant phylogeny according to the research progress of the chloroplast ycf gene family.

## DISCUSSION

The GC content of the Jerusalem artichoke IR region is high. This may be due to the fact that the IR region contained four high-GC rRNA genes (*Asaf et al., 2016*). The high G-C content made conservation in the IR regions higher than that in the LSC and SSC

regions (*Yang et al., 2014*). The sequence and composition of the chloroplast genes of the Jerusalem artichoke were similar to those of other crops of the composite family (*Curci et al., 2015*). In addition, we compared the plastid genome and the chloroplast genome of the Jerusalem artichoke. This comparison revealed that the plastid genome was 384 bp smaller than the chloroplast genome. We further refined the chloroplast genome of the Jerusalem artichoke via comparison with that produced by Bock et al. A total of 15 differentially encoded genes were found in the published Jerusalem artichoke genome sequence (*Bock et al., 2014*). These differences may be due to the differences in sequencing depth and read length between these studies, as accuracy and length of sequences from the Illumina HiSeq 2000 is less than that from the Illumina HiSeq 2500 PE150, which has 100× depth. The 95× is more refined than the genome of the plastid genome, and depth of sequencing affects the number of detected genes, as well as the statistics and expression-related downstream analyses (*Desai et al., 2013*). A paired-end sequencing approach can also lead to differences in gene detection, as for the same number of reads, paired-end $2 \times 150$ bp reads contain more information than do paired-end $2 \times 100$ bp reads (*Chaisson, Brinza & Pevzner, 2009*). In addition, we employed different genome assembly methods than did Bock et al., which may also result in differences in genome sequencing. In conclusion, a 384 bp difference in the conserved chloroplast genome may be an artifact as a consequence of the results of late cluster analysis studies, as we found that the overall difference in the chloroplasts of the Composite family ranged between 200 and 400 bp. These results will aid future chloroplast genome evolution studies and research on the positive selection of genes. Based on these sequencing results, we were able to comprehensively analyze the characteristics of the Jerusalem artichoke chloroplast genome.

Introns play an important role in selective gene splicing. Because the chloroplast genome was simple, relatively conserved and maternal, chloroplast SSR were highly efficient molecular markers. Moreover, cpSSRs have been widely used previously in crossbreeding, biogeography, and population genetics studies (*Bayly et al., 2013*). This is consistent with the chloroplast genomes of most angiosperms (*Raveendar et al., 2015*; *Yang et al., 2014*). In regards to repeat length, most SSR had 10–20 bp, while fewer had less than 10 bp, indicating that the SSR segment of the Jerusalem artichoke chloroplast genome is short. However, the long repeated sequence might promote the rearrangement of the chloroplast genome, causing an increase in population genetic diversity (*Qian et al., 2013*). This may be related to the vegetative propagation of Jerusalem artichoke, which greatly reduces the probability of genetic variation. The SSR sites distributed in the non-coding region are the majority, while only three genes in the coding region have SSR sites, and there are few SSR sites in the coding region of the chloroplast genome, as has been confirmed in Quercus and Saxifragaceae (*Liu et al., 2018*; *Yang et al., 2016*). These repetitive structures provide valuable information resources for the future development of molecular markers in the study of the phylogenetic evolution and population genetics of Jerusalem artichoke.

A comparative analysis of the coding regions in the chloroplast genome of plants in the composite family showed that Jerusalem artichoke and *H. petiolaris subsp. fallax* had the fewest differences. As a whole, the chloroplast genome of crops in the composite family

tends to be conserved. An mVISTA analysis showed that the coding region was more conserved than the non-coding region, which is consistent with reports on crops in the composite family, such as *Cynara cardunculus* (*Curci et al., 2015*) and *Ageratina adenophora* (*Nie et al., 2012*). The *ycf2* gene showed the greatest degree of differentiation. In addition, there was a gene deletion in the crops of genus *Helianthus*. Currently, many different gene regions are considered potential tools for phylogenetic analysis. These DNA domains will play an important role in the application of molecular phylogeny in this species (*Nie et al., 2012*). The *ycf2* gene is the largest known plastid gene in angiosperms (*Drescher et al., 2000b*). Although the *ycf2* gene can be used to predict phylogenetic relationships (*Drescher et al., 2000a*), its function remains unclear. This suggests that the *ycf2* gene is highly conserved in the evolution of the species within the composite family. The *ycf2* gene appears to gradually degenerate compared in gramineous crops, with only 734 bp remaining in rice and wheat (*Matsuoka et al., 2003*). The results of phylogenetic tree analysis using partial angiosperm *ycf2* genes were consistent with those obtained from the whole plastid genome data phylogenetic tree analysis. This provides even more precise details for evolutionary evaluation (*Doorduin et al., 2011*).

The composite family is one of the largest families in the plant kingdom, and the chloroplast genome plays an important role in plant classification and phylogenetic analysis. To date, abundant research has evaluated the phylogeny of crops in the composite family. Notably, study of the evolution of the *Aster spathulifolius* chloroplast genome has revealed that it bears its closest relationship with *J. vulgaris* (*Choi & Park, 2015*; *Huang, Sun & Zhang, 2010*; *Soltis et al., 2000*), which is consistent with previous reports on the uncertainty of the evolution of the Senecioninae tribe (*Doorduin et al., 2011*). In the group of the composite in which the number of involved species is more than or equal to 2, it can be seen that genetically Jerusalem artichoke is more closely related to other species of composite family, such as genus *Helianthus*. At the same time, Jerusalem artichoke is also the earliest isolated species of the genus *Helianthus*. This provides a theoretical basis for the further study of the relationship between phylogenetic branches of Jerusalem artichoke in the composite family.

The *ycf2* gene fragment is large, and the function of its open reading frame fragment is not clear. Compared with other chloroplast coding genes, the nucleotide sequence identity between *ycf2* of different families is very low, which is less than 50% in bryophytes, pteridophytes, and spermatophytes (*Wicke et al., 2011*). In the increasing number of *ycf* gene studies, although *ycf2* is highly conserved, the *ycf2* gene shows a wealth of phylogenetic information in the Orchidaceae phylogeny. *Huang, Sun & Zhang (2010)* found that the *ycf2* gene has multiple positive selection loci during angiosperm development, and the phylogenetic signal of *ycf2* probably originates from its large sequence length, so that the *ycf2* gene is valuable for future research. Most chloroplast genes were in a negative selection state in Holcoglossum, but 14 positive selection loci were detected in the *ycf2* gene (*Li et al., 2019*). In this study, some positive selection signals were found by establishing evolutionary trees of the adaptive evolution of the *ycf2* gene in the composite family, but the loci were few, which may be related to the number of species. Plants may have a variety of strategies to adapt to the environment, and adaptive modifications to other abiotic stresses of genes in the nucleus are sufficient to maintain the homeostasis of photosynthesis.

Therefore, there is no need for adaptive evolution in the chloroplast coding genes (*Dolhi, Maxwell & Morgan-Kiss, 2013*; *Kanzaki et al., 2017*; *Wang et al., 2019*). In this study, research on the *ycf2* gene in the composite family supports the idea of adaptive evolution, but there are currently few studies on adaptive evolution in Compositae crops. Therefore, further studies on the adaptive evolution of chloroplast genes in other species of the composite family are needed to explore how to adapt to these changes in environmental migration and climate change.

## CONCLUSIONS

In this study, the complete chloroplast genome sequence of Jerusalem artichoke was successfully assembled, annotated and analyzed. The chloroplast genome of the plants in the composite family is relatively conserved. Variations of the chloroplast genome are scarce between Jerusalem artichoke and plants in the same genus. Compared with composite plants belonging to other genera, we found deletions in the chloroplast genome of Jerusalem artichoke. The identification of repetitive sequences in the chloroplast genome of Jerusalem artichoke, particularly SSR, will be helpful for the development of molecular markers, the study of population genetics and the phylogenetic analysis of Jerusalem artichoke. A phylogenetic analysis of plants in the composite family shows that Jerusalem artichoke and *H. petiolaris* subsp. *fallax* share the closest relationship, both belonging to the composite family, genus *Helianthus*. The results of this study indicate *ycf2* gene has been subject to adaptive evolution, and it is suggested that more extensive investigation and in-depth discussion should be conducted in future studies. Completion of the sequencing of the chloroplast genome will provide key genetic information for further research on Jerusalem artichoke and deepen our understanding on the evolutionary history of the chloroplast genome and phylogenetic position of Jerusalem artichoke. In addition, it may be useful for various molecular biology applications of Jerusalem artichoke in the future.

### Funding

This project was supported by the National Natural Science Foundation of China (31460523, 31660588, 31660569, 31760600), the Open Project of Qinghai Key Laboratory of Qinghai-Tibet Plateau Biological Resources (2017-ZJ-Y10), and the Fundamental Research Program of Qinghai (2016-ZJ-751, 2017-ZJ-Y18). The funders had no role in study design, data collection and analysis, decision to publish, or preparation of the manuscript.

### Grant Disclosures

The following grant information was disclosed by the authors:
National Natural Science Foundation of China: 31460523, 31660588, 31660569, 31760600.
Open Project of Qinghai Key Laboratory of Qinghai-Tibet Plateau Biological Resources: 2017-ZJ-Y10.
Fundamental Research Program of Qinghai: 2016-ZJ-751, 2017-ZJ-Y18.

## Competing Interests

The authors declare that they have no competing interests.

## Author Contributions

- Qiwen Zhong performed the experiments.
- Shipeng Yang analyzed the data, prepared figures and/or tables, approved the final draft.
- Xuemei Sun prepared figures and/or tables, authored or reviewed drafts of the paper.
- Lihui Wang contributed reagents/materials/analysis tools.
- Yi Li conceived and designed the experiments.

## DNA Deposition

The following information was supplied regarding the deposition of DNA sequences:

The complete chloroplast genome of Jerusalem artichoke is available in GenBank: MG696658.

## Data Availability

Data is available at NCBI under accession number MG696658.

## Supplemental Information

Supplemental information for this article can be found online at http://dx.doi.org/10.7717/peerj.7596#supplemental-information.

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
