# Peer review of "The complete chloroplast genome of the Jerusalem artichoke (Helianthus tuberosus L.) and an adaptive evolutionary analysis of the ycf2 gene"

_PeerJ, doi:10.7717/peerj.7596_

## Round 0.1 · original submission · Major Revisions

Dear author,

Your paper has been assessed by two reviewers and myself as academic Editor.

As you could see below, the manuscript needs a major revision.
Most importantly:

Please clarify the additional contributions compared to (Bock et al., 2014)
Three plastid genomes of Helianthus tuberosus were already published at four years ago (Bock et al., 2014, New Phytologist 201: 1021-1030) and they are available from NCBI database. Does this manuscript just adds one more different sequence accession? How much different?
Please also address all concerns of the two reviewers and submit a revised version of the manuscript. Please include a detailed response to each reviewer.

Reviewer 1 ·

Basic reporting

Dear Editor,

Authors report the first chloroplast genome of Jerusalem artichoke and presents its comparison with other members of the family They have observed some differences at the ycf2 gene level. The artichoke plastome showed high synteny to the other plastomes of the same genus and clade.

Experimental design

Experimental design and methods are the standard methods.

Validity of the findings

Submitted version of the paper reports the findings on the CP genome and the authors have compared the assembled chloroplast to the other CP genome chloroplast and observed conservation. The assembled CP genome is in lengthwise similar to the other CP genomes of the same species and also the gene conservation has been observed except with the deviation in ycf2.

Additional comments

We recommend authors to look at the ycf2 functions and the surrounding regions of the ycf2. In the MS, they have mentioned that the ycf2 showed several amino acids missing and also reported deletions in the ycf2. Therefore, taking into account the recent use of the ycf2 in phylogenetic studies it would be necessary to ascertain that whether this gene or better a pseudogene can be used a phylogenetic marker for these species. Hence we recommend authors to plot the rates of the evolution along with the missing of the deletions/insertions and then calculate the correlation. If the correlation is positive at an 0.05 level or 0.01 level then it is obvious that the deletions or the observed insertions has a role to play in the selective differentiation of this gene in these species.

Reviewer 2 ·

Basic reporting

This manuscript reports the complete chloroplast genome of Jerusalem artichoke (Helianthus tuberosus) from a Chinese cultivated population. This is simple and straight forward genome report paper rather than in depth genome application paper.

Furthermore, the three plastid genomes of Helianthus tuberosus was already published at four years ago (Bock et al., 2014, New Phytologist 201: 1021-1030) and they are available from NCBI database.

Experimental design

The authors follow the well-established NGS sequencing and annotation procedures.

Validity of the findings

This manuscript reports the approximately 150 bp deletion mutation at the 5’ portion of ycf2 gene. But, the deletion is not unique in this report but share by all published Helianthus species and populations.

Additional comments

This reviewer recommends to the authors to publish the manuscript to some simple genome report journal such as Mitochondrial DNA Part B resources or other journals rather than PeerJ.

---

## Round 0.2 · Minor Revisions

Dear author,

The prior reviewers were unfortunately unavailable to re-review. As a result, we found an additional reviewer and so your paper has now been assessed by three reviewers and myself as academic Editor.

As you could see below, the manuscript has improved considerably, but the reviewer has not yet given the green light of full acceptance. They have some minor comments and also append an annotated manuscript. I think you can easily incorporate these revisions, so please submit a revised version of the manuscript.

Reviewer 3 ·

Basic reporting

The authors report on the sequencing, assembly, and annotation of the chlroplast Helianthus tuberosus L. The authors also present an evolutionary analyisis of its ycf2 gene.

The manuscript covers the sequencing and annotation of the mentioned chloroplast and compares it to other sequenced species of the same genus (Heliantus) as well as related crops, and plastids of the same species previously reported. It is a well organized manuscript but it would improve if it was thoroughly proof-read for incorrect English expressions used (I´m attaching a track-changes word document with some -but not all- of these).

To the best of my knowledge, the Introduction and background cover the relevant literature, although there are a few citations on the text that are not present on the references section.

Experimental design

There are some oversights on part of the authors regarding the availability of their data that hinder the full reproducibility of their work.

1. The authors should deposit the original short reads from their NGS experiments on an appropriate database (for example the SRA at NCBI).

2. The Multiple Sequence Alignments (MSA) used for the phylogentic trees and other analysis should be included as supplementary files as plain-text files in an appropriate format (FASTA aligned, Clustal, nexus, phylip, for example).

3. The parameters used on all bioinformatics software should be reported. If defaults were used this should be clearly stated on the corresponding methodology section. This is specially important on the assembly software, since the comparison with a previous plastid genome is noted to differ due in part to the method used.

4. The accession number for the previously reported Heliantus tuberusus plastid genome is incorrect. It reads NC023112 but should be NC_023112.

Validity of the findings

As the authors state on the first paragraph of the discussion, the differences on the sequencing technology and assembly methodology are more than likely the reason there is an 384 bp difference between the chloroplast genome presented in this manuscript and the plastid genome of Bock et al. Since the length variation on the chloroplasts of the 8 species of the composite family are in this same range (200-400 bp) I’m not so sure that the statement on lines 364 to 366 are actually valid (as this may be very well just an artifact of the different methodologies used):
“The length variations of the chloroplast genomes of 8 species of the composite family correlated with the lengths of the IR regions, indicating the length of IR region had a significant effect on the length of genome (Guo et al. 2017).”

Additional comments

Regarding the 152 amino acid sequence deletion at the ycf2 gene, it is quite interesting that is only observed on Heliantus species. Has it been corroborated by PCR or has just been observed by the assembly through NGS?

Annotated reviews are not available for download in order to protect the identity of reviewers who chose to remain anonymous.

---

## Round 0.3 · Minor Revisions

Dear author

I can read that you have addressed all the reviewers concerns. The reviewers comments have been responded adequately.

Nevertheless, the Section Editor, Julin Maloof, has made a comment. He said:

"Neutral sites are needed for proper phylogenetic reconstruction.
The Abstract states that the ycf2 gene has been subject to adaptive evolution and has the potential to be used as a phylogenetic reconstruction locus in the composite family. A similar statement is made in the conclusion and maybe elsewhere. If the gene has been subject to strong positive selection (as is shown here) then it is not appropriate to use it for phylogenetic reconstruction. "

I partially agree for a genomic/bioinformatic focus, but my personal opinion as biologist is that all traits can be used to study evolution and phylogeny. Including morphological traits. There are some advantages and disadvantages of using neutral traits, and the same is true for traits subjected to positive/negative selection.

Please clarify that a genetic locus with strong positive selection may have some advantages/disadvantages for phylogenetic and evolutionary questions.

I congratulate you for the nice piece of work, which will add value to PeerJ.

---

## Round 0.4 · accepted · Accept

Dear author

I can read that you have addressed the main concern of the section editor by removing the statement about phylogeny (in abstract and discussion).
I congratulate you for the nice piece of work, which will add value to PeerJ.